# A Systematic Review of Integrated Learning Definitions, Frameworks, and Practices in Recent Health Professions Education Literature

**Davina Matinho †, Marisa Pietrandrea †, Carlos Echeverria, Ron Helderman, Madison Masters, Daniel Regan, Samuel Shu, Rafael Moreno and Douglas McHugh ***

Frank H. Netter MD School of Medicine, Quinnipiac University, Hamden, CT 06518, USA; davina.matinho@quinnipiac.edu (D.M.); marisa.pietrandrea@quinnipiac.edu (M.P.); carlos.echeverria@quinnipiac.edu (C.E.); ron.helderman@quinnipiac.edu (R.H.); madison.masters@quinnipiac.edu (M.M.); daniel.regan@quinnipiac.edu (D.R.); samuel.shu@quinnipiac.edu (S.S.); rafael.moreno@quinnipiac.edu (R.M.)
* Correspondence: douglas.mchugh@quinnipiac.edu
† These authors contributed equally to this work as first authors.

**Abstract:** International curricular redevelopment and quality improvement efforts include integration within and across disciplines as a focal point. Definitions and applications of the term 'integration' vary routinely in health professions education literature, weakening opportunities to enhance our educational practice through collaborative, interprofessional knowledge-sharing. This systematic review examined recent health professions education literature for reported definitions, theories or frameworks, and educational activities around integrated learning, education, curricula, or teaching. A total of 35 articles representing learners from 12 different health professions, between 2017 and 2021, were analyzed through a literature search of seven databases: PubMed, Scopus, CINAHL, JSTOR, the Cochrane Library, LGBTQ + Life, and ERIC. Full-text retrieval and data extraction of the included studies were conducted. Of the 35 articles reviewed, 13 included explicit definitions of integration (an additional six alluded to a definition), 19 referred to an educational theory or framework used to guide integration design efforts, and 27 mentioned teaching methods by which integration was implemented. Misunderstanding what is meant by integrated, how others have planned for it, or how others have sought to bring it about practically, all threaten attempts to improve the cultivation of health professionals as integrated thinkers and holistic care providers.

**Keywords:** health professions education; medical education; integrated learning; integrated teaching; integrated education; integrated curriculum; horizontal integration; vertical integration; spiral curriculum

## 1. Introduction

'Integration' as an educational concept may be expressed legitimately in many different forms. For example, one form is the integration of related disciplines. Historically, health profession learners were exposed to isolated discipline-specific knowledge and skills by taking separate, distinct courses (e.g., a physiology course, a pharmacology course, a biochemistry course, etc.) before being introduced to clinical skills in later years. Critiques of this approach include it being left to learners themselves to form meaningful and relevant links between subject areas, and that a lack of cognitive scaffolding for such connected thinking leads them to struggle when called upon to apply cross-disciplinary knowledge to clinical practice [1]. Indeed, as early as 1958, Capehart observed that, "It does not always follow that several units add up to unity. In fact, a subject made up of units tends, from the viewpoint of the student, to fall into a pattern of discrete, separate and unrelated experiences [2]." The movement towards integration envisioned an alternative means to organize material to be learned and included a push to encourage learners to

think like professional practitioners from the beginning of their training [3]. For example, teaching interdisciplinary basic science knowledge within the context of clinical cases, which learners are likely to encounter during future clinical rotations and professional practice. As a consequence, the learning experience is considered to be more relevant and meaningful to students, helping them acclimatize to professional expectations and form a professional identity earlier [4].

Integration as an advantageous motif for curriculum or instructional method design has risen in esteem and international prevalence over the last half century among institutions educating health professionals [5,6]. However, while no doubt widely recognized, what is exactly meant by 'integration' by a given individual varies in the health professions education literature [5]. This divergence in shared comprehension is evident whether the term integration is applied as a descriptor to learning, education, curriculum, or teaching. The New Oxford American Dictionary defines 'integration' simply as the action or process of "combining one thing with another so that they become a whole" [7]. In 2015, Brauer and Ferguson noted of educational literature that the term integration often serves as a buzzword in the absence of unified definitions [5]. Vague or poorly expressed understandings of integration can hamper attempts to improve health professions education when it becomes a concept and a process that means different things to professors, clinicians, learners, and administrators. For example, Harden notes that discussions about integrated curricula can be quite polarizing due to, in part, how we use the word integrated [8]. Certainly, talking past one another or employing integration merely as an item of professional jargon represents missed opportunities to enhance our communities of practice—either locally or via peer-reviewed literature—through deficits in contributing ideas and strategies, crowdsourcing solutions, and learning from those with shared responsibilities. This may be especially true when 'integrated' is used as an adjective to qualify some other educational construct such as 'learning'. For example, Huber et al. (2005) described *integrated learning* as "the process of learners making connections among concepts and experiences so that information and skills can be applied to novel and complex issues or challenges" [9]. In contrast with this, Jette et al., (2004) said, "integrated learning can be separated into three categories–horizontal, vertical, and spiral" [10]; Hendriksen et al., (2020) stated that, "integrated learning involves near-peer teaching of senior students training junior students, and can be focused on experiential learning" [11]; and Dillenbourg (2004) reported that, "integrated learning refers to the organic interleaving of computerized activities (e.g., simulations, forums, exercises) with the diverse activities that occur in 'on-campus' courses (e.g., lectures, exercises, practical work, or even field trips)" [12].

This systematic review explored recent health professions education literature concerning reported definitions, theories or conceptual frameworks, and actionable educational activities around *integrated learning* and the related constructs of *education*, *curriculum*, or *teaching*. By summarizing the various points of view of health professions educators, perhaps greater common ground can be found on which to build more effective and cohesive models/methods of integrated learning.

*Research Questions*

Our three research questions were:

1. How do authors define integrated learning, integrated education, integrated curriculum, or integrated teaching?
2. What theories or conceptual frameworks are used as guidance in developing in integrated learning, education, curriculum, or teaching?
3. How do practitioners integrate learning, education, curriculum, or teaching?

## 2. Materials and Methods

### 2.1. Literature Identification

This investigation used the Preferred Reporting Items for Systematic Reviews and Meta-Analyses (PRISMA) guidelines as a design and implementation framework [13] with

three stages: (1) *Planning*: defining our major research questions; (2) *Search*: identifying appropriate literature databases, defining search terms, strategies, as well as inclusion and exclusion criteria, then subsequently conducting the literature search itself; and (3) *Literature analysis and report formulation*: article screening preceded full-text appraisal and analysis of our included studies, which was followed by data extraction and interpretation of results.

Seven electronic databases that index educationally relevant health professions literature were searched: PubMed®, Scopus®, the Cumulative Index to Nursing and Allied Health Literature (CINAHL), JSTOR, the Cochrane Library, LGBTQ + Life, and Education Resources Information Center (ERIC). The search strategy consisted of the following search terms and Boolean operators applied to the article title/abstract field in each individual database: "Integrated" AND "learning" OR "education" OR "curriculum" OR "teaching". The retrieved articles titles and abstracts were extracted and imported into the online systematic review management software, Covidence (covidence.org; Melbourne, Australia).

## 2.2. Literature Selection

The *inclusion criteria* applied were:

- articles published between 2017–2021;
- articles in English;
- adult learners;
- health professions learners from the following fields: allopathic medicine, anesthesiologist assistant, audiologist, chiropractic medicine, dentistry, dietician, genetic counseling, naturopathic medicine, nursing, nutrition, occupational therapy, optometry, orthotics, prosthetics, osteopathic medicine, pharmacy, physical therapy, physician assistant, podiatry, public health, radiation therapy, and speech pathology;
- trainees as learners (e.g., not qualified professionals or patients as learners);
- undergraduate medical education (e.g., MD degree) or other health professions equivalent (e.g., no residents); and
- peer-reviewed articles (e.g., not letters to editor, conference abstracts/presentations).

The *exclusion criteria* applied were:

- articles published before 2017;
- articles not in English;
- non-adult learners (e.g., children, adolescents);
- non-health professions learners; patients as learners;
- graduate medical education (e.g., residents) or health professions equivalent;
- already qualified professionals as learners;
- conference abstract/presentations;
- letters to editor;
- non-peer-reviewed articles;
- articles concerning integration of work with learning;
- articles concerning integration of care or management or treatment;
- articles concerning machine learning; and
- articles concerning integration as mere context (i.e., integration is not what the article is about).

The PRISMA flow diagram indicates the literature selection processes (Figure 1). All authors participated in establishing inclusion and exclusion criteria, screening the articles, full-text review, and data extraction. During the title/abstract screening, full-text analysis, and data extraction stages, each article was randomly assigned to two of the nine authors for review, with each member of the random pairings conducting their reviews independently of the other. The online systematic review management software, Covidence, was used to facilitate these processes. Covidence identified any discrepancies in consensus between independent reviews, which were resolved by the corresponding author. Our search strategy was implemented on 6 November, 2021 and identified 3897 articles from the seven databases, of which 982 duplicates were removed. After application of inclusion

and exclusion criteria to the initial title/abstract screening, 2006 articles were removed. The remaining 909 articles were assessed for eligibility based on a full-text review. Of these, 874 were excluded based on inclusion/exclusion criteria, leaving 35 articles which were included in this review [10,11,14–46]. Full-text PDFs of each of the 35 articles were imported into Covidence to facilitate data extraction.

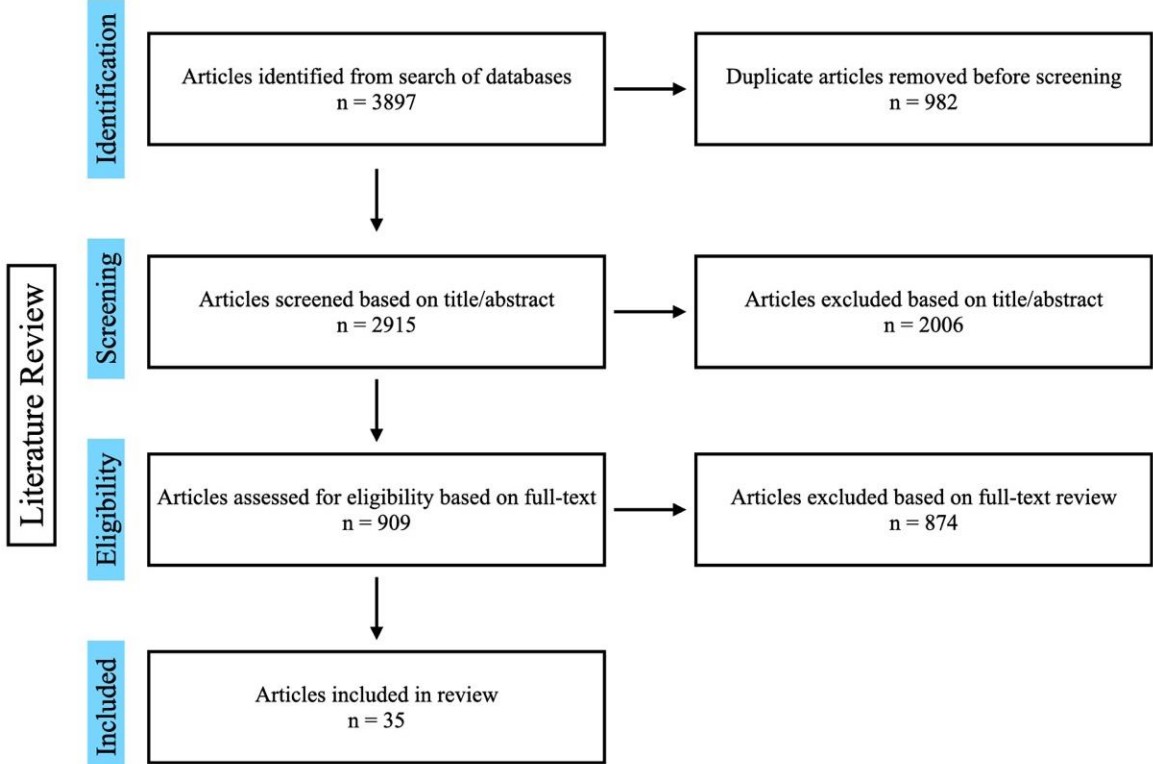

**Figure 1.** PRISMA flow diagram illustrating the literature selection processes.

### 3. Results

*3.1. Synopsis of Included Articles*

The 35 articles included in this review refer to health profession learners from sixteen countries: Antigua and Barbuda, Australia, China, France, India, Malaysia, the Netherlands, Pakistan, Russia, Saudi Arabia, South Korea, Sweden, Switzerland, the United Arab Emirates, the United Kingdom, and the United States of America (Figure 2). These articles represent 12 health professions, 28 journals, and 11 article types (Appendix A Table A1).

A summary of the 35 articles is presented regarding the major categories of author and publication year, journal, article title, location, type of health profession learner, and article type (Appendix A Table A2).

Of the 35 articles reviewed, 13 included explicit definitions of *integrated learning, education, curriculum, or teaching* (an additional six articles alluded to a definition); 19 referred to an educational theory or conceptual framework used to guide the design or development of *integrated learning, education, curriculum, or teaching*; and 27 mentioned practical methods or activities by which to implement *integrated learning, education, curriculum, or teaching* (Appendix A Tables A3–A7).

Below, we provide an overview of our findings, their relevance, and highlight a few examples associated with *integrated* definitions, theories and frameworks, and educational practices.

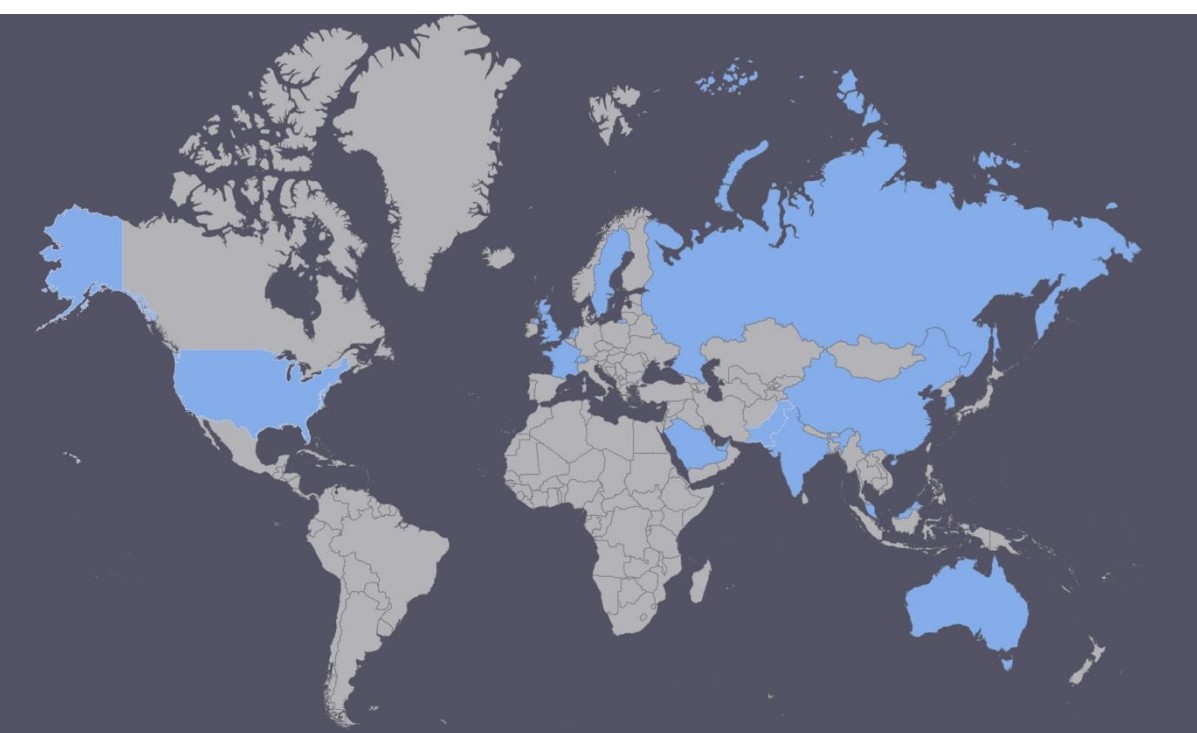

**Figure 2.** Geographical locations of the reviewed articles' learners (Antigua and Barbuda, Australia, China, France, India, Malaysia, the Netherlands, Pakistan, Russia, Saudi Arabia, South Korea, Sweden, Switzerland, the United Arab Emirates, the United Kingdom, the United States of America).

*3.2. Definitions of Integrated Learning, Education, Curriculum, or Teaching*

In percentage terms, ~37% of the reviewed articles stated clearly and in detail what they meant by *integrated*. Of these, authors' elucidations fell largely into three major themes: (1) *the extent to which knowledge, beliefs, or skills are co-presented by educators to learners* [47], such as the teaching of insulin-related biochemistry, physiology, and pharmacology in the context of diabetes mellitus and wider public health disparities around prediabetes; (2) *an organizational approach that informs how curricular elements are structured and arranged* [5,48,49], such as horizontal, vertical, or spiral integration; interdisciplinary organ-system blocks; or longitudinal integrated clerkships; and (3) *cognitive or metacognitive processes occurring within the learner's mind* [1], such as students establishing significant linkages between different subject areas and skill sets through independent and collaborative thinking so that they can understand, critically appraise, and apply knowledge or skills to novel situations.

The first of these themes is illustrated well by Harden's integration ladder and its 11 steps [8]. Harden presented a ladder with 11 rungs as a model to aid in planning, implementing, and evaluating integration efforts. Steps 1–11 (1 = "bottom of ladder" and least integrated; 11 = "top of ladder" and most integrated) are as follows:

1.  Isolation,
2.  Awareness,
3.  Harmonization,
4.  Nesting,
5.  Temporal Coordination,
6.  Sharing,
7.  Correlation,
8.  Complementary,
9.  Multidisciplinary,
10. Interdisciplinary, and
11. Transdisciplinary.

Each individual rung and the progression from one to the next are clearly outlined so that educators can use this model to guide their own thinking around integration. Towards the bottom, the teaching focus is more on separate disciplines; towards the top, the teaching focus concerns high-fidelity learning where students integrate all pertinent subject areas and skill sets intrinsically, and achieve authentic competence with real word tasks [8]. When authors communicated with this intended definition, they used words or phrases such as: "incorporate humanities or the social sciences", "different levels", "cross-disciplinary", "basic science contextualized with clinical cases", "breakdown barriers", "remove silos"—which were associated with different rungs on the Harden ladder.

The second theme is exemplified by references to horizontal, vertical, or spiral integration. *Horizontal* integration concerns integration between different subject areas within a limited time period [5]. For example, individual courses can be incorporated into interdisciplinary blocks prior to clinical learning [48]. *Vertical* integration refers to integration across time to improve the connection between the basic and clinical sciences [5]. Tactics used by some U.S. medical schools to integrate basic science into core clerkships (e.g., internal medicine, surgery, obstetrics and gynecology, pediatrics, family medicine, psychiatry, neurology) include transition-to-clerkship courses, longitudinal sessions reinforcing the foundational sciences during clerkships, asynchronous online case-based learning, and bedside basic science teaching scripts [50]. On the other hand, *spiral* integration involves integration across disciplines and time, such that learners revisit material at increasing levels of complexity as they progress through the course [50]. For example, in a spiral nursing curriculum, a *first exposure* to oxygenation may involve teaching students the purpose of oxygen in a patient's health, how to assess whether the patient's oxygenation in their blood and body is optimal, and discussing different methods to improve oxygenation. There may be a focus on oxygen administration devices, how a nurse cares for patients with these devices, and the need for the nurse to assess that the devices are working. Later, a *second exposure* may build upon the concept of oxygenation by bringing in some disease processes related to oxygenation (e.g., upper respiratory diseases, chronic obstructive pulmonary disease, lung cancer, pneumonia). Later again, a *third exposure* may cover medical-surgical scenarios. The nursing student could be introduced to a complex patient with multiple problems and expected to independently apply the nursing process and critical thinking in order to "put it all together" (i.e., integrate).

The third theme is anchored in the idea that integration " . . . happens inside the head of the individual learner, through the combination of prior knowledge with new information and/or experiences [1]". The educator's role then becomes one where they (1) create and facilitate opportunities for students to form relevant interconnected cognitive links, and (2) support students' self-regulated learning processes through formative assessment and feedback. Self-regulated learning concerns, "self-initiated and self-monitored activities, practices, and behaviors that learners engage in to pursue academic mastery." [51]. When authors communicated with this intended definition, they used words or phrases such as: "exercise independent and collaborative thinking", "experiential", "scaffold for learners", "connect new meanings", "connect new understandings between knowledge", "respond to problems by combining knowledge and skills", "assimilate and apply", and "learning outputs include: understand, evaluation, manage".

*3.3. Theories and Frameworks for Integrated Learning, Education, Curriculum, or Teaching*

In percentage terms, ~54% of the reviewed articles made a straightforward reference to an educational theory or conceptual framework they considered helpful to guide the development of integrated learning, education, curriculum, or teaching. A total of 22 distinct theories or frameworks were identified (Appendix A Table A6):

- Blended learning theory;
- Blending with pedagogical purpose model;
- Bloom's taxonomy;
- Cognitive flexibility theory;

- Cognitive load theory;
- Community of inquiry framework;
- Connectivism;
- Constructivism;
- Gagne's cognitivist instructional design;
- Glassick's educational scholarship criteria;
- Horizontal integration, vertical integration, spiral curriculum;
- Interactive, constructive, active, passive (ICAP) framework;
- Intervention mapping framework;
- Kern's six steps for curriculum development;
- Multimodal model for online education;
- Online collaborative learning;
- Self-determination theory;
- Self-regulated learning theory;
- The nutrition care process (NCP);
- The rubric of Medical/Dental Humanities-Social Medicine/Dentistry (MDHS) education;
- Thistlethwaite and Nisbet's markers of quality inter-professional education [52];
- The Association of American Medical Colleges (AAMC) Advisory Committee on Sexual Orientation, Gender Identity, and Sex Development issued a competency-based report, "Implementing Curricular and Institutional Climate Changes to Improve Health Care for Individuals who are LGBT, Gender Nonconforming, or Born with DSD: A Resource for Medical Educators" [19]; and
- "A theoretical model from the management literature can help frame how to integrate the core competencies for interprofessional practice with uni-professional curricula . . . educators need to provide multiple settings and configurations for learners to hone their collaborative skills so that they become an unquestioned part of their practice." [29].

Educational theories and frameworks enable crosstalk between what could otherwise be educational research silos [53]. Theories help explain the application, interpretation, and purpose of learning and education [54], whereas frameworks are research-informed models that help educators to understand the why and how of a particular phenomenon and to guide their instructional design efforts [55]. Both have the potential to inform educational approaches to integration at the level of individual teaching events and whole curricula, and, when included in scholarly publications, they facilitate knowledge-building conversations between learning research and educational practice [56,57].

One of theories referenced in the articles reviewed is *constructivism* (Appendix A, Table A6). This is a theory of learning that says people construct their own understanding and knowledge of the world through experiencing things and reflecting on those experiences. When we encounter something new, we have to reconcile it with our previous ideas and experience, maybe changing what we believe, or maybe discarding the new information as irrelevant. In any case, we are active creators of our own knowledge. To do this, we must ask questions, explore, and assess what we know. Put succinctly, it posits that learning is an active, contextualized process of constructing knowledge through experience rather than acquiring it [57]. In contrast with this, a more passive approach to teaching and learning includes the 19th or 20th century traditional lecture. In this paradigm, effective teaching involves the efficient and effective transmission of a specialized body of knowledge and way of thinking (e.g., subject matter expertise in speech pathology), similar to what is possessed by the teacher or contained in a textbook. Fink (2013) writes that, "a long history of research indicates lecturing has limited effectiveness in helping students retain information after a course is over, develop an ability to transfer knowledge to novel situations, develop skills in thinking or problem solving, and achieve affective outcomes, such as motivation for additional learning or a change in attitude [58]."—all of which may be reasonably argued are important to integration. Another relevant example here is the ICAP framework referenced in another of the reviewed articles (Appendix A Table A6). The ICAP framework for cognitive engagement can be used to categorize educational pedagogies as *interactive*,

*constructive*, *active*, or *passive* [59]. Quesnelle et al., (2021) used the ICAP framework to investigate the level of cognitive engagement required of medical students in standalone pharmacology learning events versus integrated sessions that included pharmacology [45]. This framework helps describe and categorize pedagogies relevant to different levels of cognitive engagement: constructive pedagogies generate new knowledge through inferring, comparing, and contrasting; interactive pedagogies involve learners "co-inferring" with peers to develop knowledge that neither partner knew previously; active pedagogies involve manipulating information where existing knowledge is integrated and emphasized; and passive pedagogies involve isolated storing of information [45].

To sum up, educational theories and frameworks pertinent to *integrated learning*, *education*, *curriculum*, or *teaching* assist educators in translating principles of learning into optimal instructional actions.

### 3.4. Actionable Practices to Achieve Integrated Learning, Education, Curriculum, or Teaching

In percentage terms, ~77% of the reviewed articles reported the methods or activities by which they went about achieving integrated learning, education, curriculum, or teaching. A total of 43 distinct practices were identified (Appendix A Table A7):

- Art-viewing and reflective question prompts;
- Bedside teaching;
- Case-based learning;
- Classroom instruction and activities;
- Clinical case orientation;
- Clinical shadowing;
- Clinical skills are chained with basic medical sciences through simulated skills in arranged topics;
- Community activities;
- Embed basic science topics within clinical problems;
- Enquiry-based learning (EBL);
- Field visits;
- Flipped classroom;
- In-class practical exercises;
- In-person or video-based lectures;
- Including inclusive and affirmative content;
- Integrated assessment;
- Integrated co-teaching;
- Integrated professional practice (IPP);
- Interprofessional education (IPE);
- Massive open online courses (MOOCs);
- Observed live patient case rounds conducted by experienced clinicians;
- Online interactive videos with built-in questions;
- Patient panel;
- Peer teaching;
- Planned integrated clinical experiences, skill checkoffs, and drill downs;
- Practical laboratory sessions;
- Problem-based learning (PBL);
- Reflective writing;
- Self-directed learning;
- Seminars;
- Service learning;
- Simulation;
- Skill laboratory sessions;
- Small group chalk talk;
- Small-group discussions;
- Spaced education;

- Standardized patients;
- Student-directed online learning activities;
- Supplemental reading;
- Task-based learning;
- Team-based learning;
- Tutorials; and
- Whiteboard lectures.

Increasing complexity and expectations around the integration of basic science and clinical teaching or practice make informed choices of teaching methods and associated assessments critical—particularly as educators and health profession institutions seek to make the best use of resources. Thoughtful consideration of what to do and how to do it matters because well-chosen teaching methods help people learn better. Instructional design is the practice of, "creating instructional experiences which make the acquisition of knowledge and skill more efficient, effective, and appealing" [60]. Many of the articles in this review described the teaching methods (i.e., actionable educational practices to facilitate human learning and development) they designed or selected in pursuit of *integrated learning*, *education*, *curriculum*, or *teaching*. The rationale for their selection of a given instructional approach often began with needs-based assessments of current curricula or an analysis of curricular content, both horizontally and vertically, by a committee of relevant stakeholders. Shared planning between educators from different elements or phases of a curriculum is increasingly important as greater degrees of integration are sought [8]. Once the relevant needs or themes were identified by authors, various practical learning strategies could be selected and implemented. Some authors focused on designing curricula to highlight relevant longitudinal themes for learners, while others used combinations of teaching methods such as (but not limited to) problem-based learning (PBL), case-based learning (CBL), enquiry-based learning (EBL), peer-teaching, integrated co-teaching, interactive modules, flipped classrooms, and traditional lectures to allow students themselves to uncover relationships between subject matter content.

PBL is an example of a teaching method that applies the key principles of the educational theory, constructivism. It is closely linked to learning through interactions with other people (i.e., social learning). Implementation of PBL involves learners working in groups to figure out a solution to a complex, real-world problem based on their own understanding of the world and the topic. For example, an endocrine/liver/gastrointestinal case may open with, "*A 45-year-old woman presents to her primary care physician's office concerning of 'pain in the middle of my belly'*". Students may use this case to stimulate their learning of cholestasis or steatohepatitis through finding explanations for the source, distribution, and underlying physiological process of the pain as they are considering her unfolding history of past illness, review of systems, past medical history, past surgical history, social history, family history, etc. In other words, during PBL sessions, learners collaborate together in teams—integrating knowledge, theory, and practice in the process. PBL is widely used around the world in multiple disciplines, and is generally well-accepted by learners, faculties, and institutions [61].

Sharma et al., (2017) selected co-teaching as an instructional format to achieve *integrated learning* [18]. This amounted to a pair of educators teaching biochemistry and general medicine together in the context of diabetes mellitus and then alcohol-related liver disease. They observed student preference for integrated co-teaching over compartmentalized teaching of the same material. This aligns with the *sharing* rung of Harden's ladder [8], where two disciplines agree to plan and jointly implement a teaching program. Thus, the overlapping concepts, insights, and relevance emerge to learners.

Cahn et al., (2018) chained classroom, simulation, clinical, and community-based learning events together to foster *integrated interprofessional education* with medical, nursing, occupational therapy, physical therapy, physician assistant, and speech-language pathology students [29]. Chaining is a teaching method where knowledge and skills are reinforced in a sequence to enable the learner to perform more complex behaviors. For the *classroom*

element, students were placed in interdisciplinary teams of 5–6 learners and were required to take a course teaching collaborative practice competencies; the *simulation* element involved each team participating in two simulated clinical scenarios with standardized patient actors; for the *clinical* element, students paired with a peer from another profession, engaged in active observation, and facilitated reflection around collaborative care in a hospital cardiac step-down unit; and the *community* element had learners read the novel "Inside the O'Briens" by Lisa Genova to confront the racial dynamics of Boston, the ethics of degenerative disease, and the possibilities for interprofessional care. Later, in their teams, students partnered with a local non-profit organization for a day of service learning.

## 4. Discussion

As we approach the mid-21st century, there remains a continuing emphasis on the need for *integrated learning* in health professions education and clinical training. Educational leaders wish to produce future practitioners who are integrated thinkers and doers. In other words, future practitioners who are adept and confident at interacting with, applying, and communicating basic science knowledge during the clinical care of patients. Indeed, advances in lifestyle and personalized medicine in the 21st century will depend on health profession clinicians who can cross-link and unite basic and clinical sciences in a manner that is personally meaningful and professionally useful.

Our review found in the majority of the health professions education articles we studied that *integrated learning, education, curriculum, or teaching* as clearly stated and well-defined concepts were absent. This replicates the observation of Brauer and Ferguson who noted the phenomenon of integration presenting as a buzzword in the absence of unified definitions in educational literature [5]. When present, definitions clustered thematically around *co-presentation*, *organization*, or *cognition* as conceptual focal points. Authors' may be assuming that their readers have the same comprehension as themselves and, therefore, do not need to concretely communicate what they mean. However, misunderstanding, talking past one another, or employing *integrated* merely as an item of educational jargon may hamper attempts to improve the cultivation of health professionals as integrated thinkers. Faculty and student stakeholders alike are vulnerable to this, given the various forms that *integrated learning* may take and the potential to hold, consciously or unconsciously, a different conceptual understanding of *integrated* than someone else. Our findings relate directly to the best practice standard of *operationalization*. With regard to educational research involving constructs (i.e., intangible abstract ideas or phenomena) such as integration, it is important to precisely define what is being investigated and subsequently reported. Without transparent and specific operational definitions, researchers may measure irrelevant concepts or inconsistently apply methods. Operationalization reduces the subjectivity and increases the reliability of studies [62]. Being attentive to how well we communicate definitions of *integrated learning*, *education*, *curriculum*, or *teaching* will enhance our communities of practice—either locally or via peer-reviewed literature.

Slightly less than half of the articles we reviewed did not make a straightforward reference to an educational theory or framework. Does this really matter? Why are educational theories or frameworks of value when it comes to designing or implementing *integrated learning*? To answer these questions, we should begin with the understanding that educational research is concerned with investigating problems relating to general questions about learning, teaching, and education that are studied in local contexts [63]. Curry et al., (2009) defined theory as, "a set of general, modifiable propositions that help explain, predict, and interpret events or phenomena of interest" [64]. In light of this, studies that acknowledge and connect to theoretical frameworks are able to enter into a knowledge-building conversation with other scholarly works [65]. This is because theory-informed frameworks provide a systematic structure and organization to support the rationale for individual studies. They justify why and how a study will be undertaken by facilitating: (1) the transformation of a personal idea or a local, concrete question into a researchable problem of general interest; (2) the formulation of a refined, focused research question;

(3) the choice of an appropriate research method; and (4) a discussion of the generalizability of the study's findings [63].

Almost three-quarters of the articles reviewed reported the practical means or methods they used to implement *integrated learning*, *education*, *curriculum*, or *teaching*. These practical methods represent strategies, pedagogy, and instruction styles used within classrooms or other learning environments. Reporting them matters from a shared professional development perspective. There is a great need to align teaching methods of a health profession faculty with individual and institutional objectives around *integrated learning*. It should also be acknowledged that the quality improvement around accomplishing integration is also occurring in a broader contextual trend in health professions education that emphasizes: (1) problem-based, student-directed, and peer-assisted horizontal collaborative learning methods [65] and (2) a transformation from the traditional authoritative role of teachers to more supervisory and mentoring conventions [66]. The former is likely to facilitate opportunities for improved *integrated learning*, while the latter may impede positive change if traditional educators are uneasy with *integrated teaching* methods, preferring to stay entrenched with discipline-specific didactic lecturing. Ultimately, the consideration and choice of specific teaching methods congruent with intended *integrated learning* outcomes is intrinsically linked to how successfully health profession students will transfer their learning to the workplace.

This review possesses multiple strengths. Firstly, we utilized seven databases to search for health professions education articles pertaining to integration. This allowed us to identify a varied selection of articles in terms of journals, location of learners, and health professional careers. Secondly, we only included peer-reviewed articles, which increased the credibility of our findings. Thirdly, we used the Covidence software for each step of the review, which allowed us to organize and collaborate effectively while ensuring accuracy via consensus. Limitations of this review include it only pertaining to articles available in English that were published between 2017 and 2021 and were concerning undergraduate medical education or its equivalent in other health professions, all of which could have caused us to miss out on data from other worthy sources.

## 5. Conclusions

*Integrated learning* in health professions fundamentally involves the ability to link concepts from different-but-related fields, engage in higher-order thinking, and apply them in response to clinical problems that impact patient care. Integration as a broader concept is complex and can take many forms—as reflected in the segmentation of shared understandings of what the term *integrated* actually means to published educational scholars, as well as the diverse set of theoretical frameworks and teaching methods they delineated. In contrast, our investigation also found that many other authors of peer-reviewed education articles concerning the health professions did not operationalize the term *integrated*, report how they connected their work to a relevant theoretical framework, or share how they pursued *integrated learning* practically—all of which represent missed opportunities with regard to best practice standards and hinder the replication and extension of their studies by others.

Future research on this topic should pay attention to post-initial-professional-degree training (e.g., graduate medical education or its equivalent) and integrated assessment (i.e., eliciting evidence of integrated learning or integrated thinking from trainees).

We hope that health profession educators and integrated curriculum developers will benefit from this review, calling attention to recent international definitions of *integrated learning*, *education*, *curriculum*, and *teaching*; theories and frameworks used to guide its development; and specific teaching methods chosen for its implementation.

**Author Contributions:** Conceptualization, D.M. (Douglas McHugh); methodology, all authors; software, D.M. (Douglas McHugh); formal analysis, all authors; investigation, all authors; resources, D.M. (Douglas McHugh); data curation, D.M. (Douglas McHugh); writing—original draft preparation, all authors; writing—review and editing, all authors; visualization, D.M. (Douglas McHugh); super-

vision, D.M. (Douglas McHugh); project administration, D.M. (Douglas McHugh). D.M. (Davina Matinho) and M.P. contributed equally to this paper as first authors. All authors have read and agreed to the published version of the manuscript.

**Funding:** This research received no external funding.

**Institutional Review Board Statement:** The institutional review board of [Blinded for peer-review per journal requirements] declared this systematic review to be exempt from federal regulations because it uses publicly available non-identifiable data and, therefore, does not meet the definition of human subjects research (45 CFR 46.110) (protocol number #00422).

**Informed Consent Statement:** Not applicable.

**Data Availability Statement:** The data presented in this study are available on request from the corresponding author.

**Conflicts of Interest:** The authors declare no conflict of interest.

## Appendix A

**Table A1.** Summary of the health professions, journals, and article types represented by the 35 articles included in this review.

| Health Professions Represented | Journals | Article Types (How Many) |
|---|---|---|
| • Clinical Nutrition and Dietetics<br>• Clinical Psychology<br>• Dental Therapy and Hygiene<br>• Dentistry<br>• Medicine (Physicians)<br>• Nursing<br>• Occupational Therapy<br>• Pharmacy<br>• Physical Therapy<br>• Physician Assistant<br>• Public Health<br>• Speech Language and Pathology | • Advances in Medical Education and Practice<br>• BMC Medical Education<br>• BMJ Open<br>• British Dental Journal<br>• Currents in Pharmacy Teaching and Learning<br>• Folia Phoniatricia et Logopaedica<br>• Frontiers in Public Health<br>• Journal of Clinical Imaging Science<br>• Journal of Interprofessional Care<br>• Journal of Physical Therapy Education<br>• Journal of Physician Assistant Education<br>• Journal of Taibah University Medical Sciences<br>• Journal of the National Medical Association<br>• Journal on Excellence in College Teaching<br>• MedEdPortal<br>• Medical Education Online<br>• Medical Forum Monthly<br>• Medical Principles and Practice<br>• Medical Science Educator<br>• Medical Science Educator<br>• Online Learning<br>• Pakistan Journal of Medical Sciences<br>• Pharmacology Research Perspectives<br>• Russian Open Medical Journal<br>• TechTrends | • Case report: 1<br>• Cohort study: 2<br>• Comparative study: 1<br>• Cross-sectional study: 1<br>• Method/model description and evaluation: 12<br>• Method/model description: 7<br>• Mixed methods research protocol: 1<br>• Monograph: 4<br>• Non-random controlled trial: 2<br>• Qualitative research: 2<br>• Random controlled trial: 2 |

**Table A2.** Summary of included articles in this systematic review. N/S = none specified.

| Reference | Authors/Year | Journal | Article Title | Location | Learners | Article Type |
|---|---|---|---|---|---|---|
| [14] | Bernauer and Fuller (2017) | Journal on Excellence in College Teaching | "Beyond Measurement Driven Instruction: Achieving Deep Learning Based on Constructivist Learning Theory, Integrated Assessment, and a Flipped Classroom Approach." | The United States of America | N/S | Monograph |
| [15] | Picciano (2017) | Online Learning | "Theories and Frameworks for Online Education: Seeking an Integrated Model." | The United States of America | N/S | Method/model description |
| [16] | Myers and Schenkman (2017) | Journal of Physical Therapy Education | "Utilizing a Curriculum Development Process to Design and Implement a New Integrated Clinical Education Experience." | The United States of America | Physical Therapy students | Method/model description and evaluation |
| [17] | Moran Tovin et al., (2017) | Journal of Physical Therapy Education | "Pediatric Integrated Clinical Experiences: Enhancing Learning Through a Series of Clinical Exposures." | The United States of America | Physical Therapy students | Method/model description and evaluation |
| [18] | Sharma et al., (2017) | Journal of the National Medical Association | "Co-teaching: exploring an Alternative for Integrated Curriculum." | India | Medical students | Randomized controlled trial |
| [19] | Holthauser et al., (2017) | Medical Science Educator | "eQuality: a Process Model to Develop an Integrated, Comprehensive Medical Education Curriculum for LGBT, Gender Nonconforming, and DSD Health." | The United States of America | Medical students | Monograph |
| [20] | DeBate et al., (2017) | Frontiers in Public Health | "Application of the Intervention Mapping Framework to Develop an Integrated Twenty-First Century Core Curriculum-Part 1: Mobilizing the Community to Revise the Masters of Public Health Core Competencies." | The United States of America | Masters of Public Health students | Method/model description |
| [21] | Corvin et al., (2017) | Frontiers in Public Health | "Application of the Intervention Mapping Framework to Develop an Integrated Twenty-first Century Core Curriculum-Part Two: Translation of MPH Core Competencies into an Integrated Theory-Based Core Curriculum." | The United States of America | Masters of Public Health students | Method/model description |
| [22] | Tshibwabwa et al., (2017) | Journal of Clinical Imaging Science | "An Integrated Interactive-Spaced Education Radiology Curriculum for Preclinical Students." | Antigua and Barbuda | Medical students | Non-randomized Experimental study |
| [23] | Baker et al., (2017) | BMC Medical Education | "Using National Health Care Databases and Problem-Based Practice Analysis to Inform Integrated Curriculum Development." | The United States of America | Medical students | Monograph |
| [24] | Carvour et al., (2018) | Medical Science Educator | "Development of an Integrated Evidence-Based Medicine Curriculum Using a Cascade Model." | The United States of America | Medical students | Method/model description and evaluation |
| [25] | Mawdsley and Willis (2018) | Currents in Pharmacy Teaching and Learning | "Exploring an integrated curriculum in pharmacy: Educators' perspectives." | The United Kingdom | Pharmacy students | Qualitative research |

**Table A2.** *Cont.*

| Reference | Authors/Year | Journal | Article Title | Location | Learners | Article Type |
|---|---|---|---|---|---|---|
| [26] | Akram et al., (2018) | Pakistan Journal of Medical Sciences | "An approach for developing integrated undergraduate medical curriculum." | Pakistan, Malaysia, Saudi Arabia | Medical students | Method/model description |
| [27] | Atta and AlQahtani (2018) | Advances in Medical Education and Practice | "Mapping of pathology curriculum as quadriphasic model in an integrated medical school: how to put into practice?" | Saudi Arabia | Medical students | Method/model description and evaluation |
| [28] | Yue et al., (2018) | BMC Medical Education | "Using integrated problem- and lecture-based learning teaching modes for imaging diagnosis education." | China | Medical students | Randomized controlled trial |
| [29] | Cahn et al., (2018) | Journal of Inter-professional Care | "Competent in any context: An integrated model of interprofessional education." | The United States of America | Medical, nursing, occupational therapy, physical therapy, physician assistant, and speech-language pathology students | Method/model description |
| [30] | Gustin et al., (2018) | Medical Education Online | "Integrated problem-based learning versus lectures: a path analysis modelling of the relationships between educational context and learning approaches." | France, Switzerland | Medical students | Cohort study |
| [31] | Zumwalt and Dominguez (2019) | Medical Science Educator | "Integrating the Educators: Outcomes of a Pilot Program to Prime Basic Science Medical Educators for Success in Integrated Curricula." | The United States of America | Advanced PhD basic science trainees planning on medical education careers. | Case report |
| [32] | Al-Nimr et al., (2019) | Medical Science Educator | "A 4-Year Integrated Nutrition Curriculum for Medical Student Education." | The United States of America | Medical students | Method/model description |
| [33] | Mawdsley and Willis (2019) | Currents in Pharmacy Teaching and Learning | "Exploring an integrated curriculum in pharmacy: Students' perspectives on the experienced curriculum and pedagogies supporting integrative learning." | The United Kingdom | Pharmacy students | Qualitative research |

**Table A2.** *Cont.*

| Reference | Authors/Year | Journal | Article Title | Location | Learners | Article Type |
|---|---|---|---|---|---|---|
| [34] | McIlwaine et al., (2019) | British Dental Journal | "A novel, integrated curriculum for dental hygiene-therapists and dentists." | The United Kingdom | Dental students, dental therapy and hygiene students | Method/model description and evaluation |
| [10] | Jette et al., (2020) | Journal of Physcial Therapy Education | "A Theoretical Framework and Process for Implementing a Spiral Integrated Curriculum in a Physical Therapist Education Program." | The United States of America | Physical therapy students | Method/model description and evaluation |
| [35] | Fatima et al., (2020) | Medical Forum Monthly | "Challenges and difficulties associated with physiology learning in undergraduate medical students in integrated curriculum." | Pakistan | Medical students | Cross-sectional study |
| [36] | Kapitonova et al., (2020) | Russian Open Medical Journal | "Is it time for transition from the subject-based to the integrated preclinical medical curriculum?" | Russia | Medical students | Monograph |
| [37] | Gergen et al., (2020) | MedEdPortal | "Integrated Critical Care Curriculum for the Third-Year Internal Medicine Clerkship." | The United States of America | Medical students | Method/model description and evaluation |
| [38] | Hendriks et al., (2020) | BMJ Open | "Uncovering motivation and self-regulated learning skills in integrated medical MOOC learning: a mixed methods research protocol." | The Netherlands | Medical students | Mixed methods research protocol |
| [11] | Hendriksen et al., (2020) | Currents in Pharmacy Teaching and Learning | "Complex patient cases solved by near-peer integrated teams provides leadership, professionalism, and peer-teaching opportunities." | The United States of America | Pharmacy students | Method/model description and evaluation |
| [39] | Banning et al., (2020) | Journal of Physician Assistant Education | "Qualitative Assessment of Arts-Integrated Education for Physician Assistant Students." | The United States of America | Physician Assistant students | Method/model description and evaluation |
| [40] | Lee et al., (2020) | BMC Medical Education | "An integrated humanities-social sciences course in health sciences education: proposed design, effectiveness, and associated factors." | South Korea | Dental students | Method/model description and evaluation |
| [41] | Venkatesh et al., (2020) | Medical Principles and Practice | "Factors Influencing Medical Students' Experiences and Satisfaction with Blended Integrated E-Learning." | Australia | Medical students | Cohort study |

**Table A2.** *Cont.*

| Reference | Authors/Year | Journal | Article Title | Location | Learners | Article Type |
|---|---|---|---|---|---|---|
| [42] | Strömbergsson et al., (2020) | Folia Phoniatrica et Logopaedica | "Towards an Integrated Curriculum in a Speech and Language Pathology Education Programme: Development and Constituents' Initial Responses." | Sweden | Speech and Language Pathology students | Method/model description and evaluation |
| [43] | Abu Farha et al., (2021) | Journal of Taibah University Medical Sciences | "Introducing integrated case-based learning to clinical nutrition training and evaluating students' learning performance." | The United Arab Emirates | Clinical Nutrition and Dietetics students | Non-randomized experimental study |
| [44] | Malhotra et al., (2021) | Journal of Inter-professional Care | "Application of constructivism and cognitive flexibility theory to build a Comprehensive, Integrated, Multimodal Interprofessional Education and Practice (CIM-IPEP) program." | The United States of America | Pharmacy, Medicine, Psychology, and Nursing students | Method/model description |
| [45] | Quesnelle et al., (2021) | Pharmacology Research Perspectives | "Design of a foundational sciences curriculum: Applying the ICAP framework to pharmacology education in integrated medical curricula." | The United States of America | Medical students | Comparative study |
| [46] | Parrish et al., (2021) | TechTrends | "Fostering Cognitive Presence, Social Presence and Teaching Presence with Integrated Online-Team-Based Learning." | The United States of America | Graduate students seeking initial secondary teacher certification | Method/model description and evaluation |

**Table A3.** Summary of the articles in this systematic review that defined integrated learning, cited a framework to guide the development of integrated learning, or described actionable educational practices intended to achieve integrated learning. N/S = none specified; X = not included; alludes = not explicitly included but the authors indirectly implied something pertinent.

| Reference | Authors and Year | Learners | Integrated Learning Definition Included | Integrated Framework Included | Integrated Practices Included |
|---|---|---|---|---|---|
| [14] | Bernauer and Fuller (2017) | N/S | X | YES | YES |
| [15] | Picciano (2017) | N/S | X | YES | X |
| [16] | Myers and Schenkman (2017) | Physical Therapy students | YES | YES | YES |
| [17] | Moran Tovin et al., (2017) | Physical Therapy students | X | X | YES |
| [18] | Sharma et al., (2017) | Medical students | X | X | YES |
| [19] | Holthauser et al., (2017) | Medical students | X | YES | YES |
| [20] | DeBate et al., (2017) | Masters of Public Health students | X | YES | X |

**Table A3.** *Cont.*

| Reference | Authors and Year | Learners | Integrated Learning Definition Included | Integrated Framework Included | Integrated Practices Included |
|---|---|---|---|---|---|
| [21] | Corvin et al., (2017) | Masters of Public Health students | Alludes | YES | YES |
| [22] | Tshibwabwa et al., (2017) | Medical students | X | X | YES |
| [23] | Baker et al., (2017) | Medical students | YES | X | YES |
| [24] | Carvour et al., (2018) | Medical students | Alludes | X | YES |
| [25] | Mawdsley and Willis (2018) | Pharmacy students | YES | X | X |
| [26] | Akram et al., (2018) | Medical students | YES | YES | YES |
| [27] | Atta and AlQahtani (2018) | Medical students | YES | X | YES |
| [28] | Yue et al., (2018) | Medical students | X | X | YES |
| [29] | Cahn et al., (2018) | Medical, nursing, occupational therapy, physical therapy, physician assistant, and speech-language pathology students | X | YES | YES |
| [30] | Gustin et al., (2018) | Medical students | Alludes | YES | YES |
| [31] | Zumwalt and Dominguez (2019) | Advanced PhD basic science trainees planning on medical education careers. | YES | X | YES |
| [32] | Al-Nimr et al., (2019) | Medical students | X | X | X |
| [33] | Mawdsley and Willis (2019) | Pharmacy students | YES | X | YES |
| [34] | McIlwaine et al., (2019) | Dental students, dental therapy and hygiene students | X | YES | YES |
| [10] | Jette et al., (2020) | Physical therapy students | YES | YES | YES |
| [35] | Fatima et al., (2020) | Medical students | Alludes | X | X |
| [36] | Kapitonova et al., (2020) | Medical students | YES | YES | YES |
| [37] | Gergen et al., (2020) | Medical students | X | X | YES |
| [38] | Hendriks et al., (2020) | Medical students | X | YES | YES |
| [11] | Hendriksen et al., (2020) | Pharmacy students | YES | X | YES |
| [39] | Banning et al., (2020) | Physician Assistant students | X | X | YES |

Table A3. *Cont.*

| Reference | Authors and Year | Learners | Integrated Learning Definition Included | Integrated Framework Included | Integrated Practices Included |
|---|---|---|---|---|---|
| [40] | Lee et al., (2020) | Dental students | YES | YES | X |
| [41] | Venkatesh et al., (2020) | Medical students | X | X | YES |
| [42] | Strömbergsson et al., (2020) | Speech and Language Pathology students | Alludes | YES | X |
| [43] | Abu Farha et al., (2021) | Clinical Nutrition and Dietetics students | Alludes | YES | YES |
| [44] | Malhotra et al., (2021) | Pharmacy, Medicine, Psychology, and Nursing students | X | YES | YES |
| [45] | Quesnelle et al., (2021) | Medical students | YES | YES | X |
| [46] | Parrish et al., (2021) | Graduate students seeking initial secondary teacher certification | YES | YES | YES |

**Table A4.** Summary of the explicit definitions of integrated learning, education, curriculum, or teaching. N/S = none specified.

| Reference | Authors/Year | Learners | Authors' Explicit Definition of Integrated Learning, Education, Curriculum, or Teaching |
|---|---|---|---|
| [15] | Picciano (2017) | N/S | "An integrated model of online education is one that provides the learner access to an educational experience that is flexible in time and space, incorporating independent and collaborative learning. Integrated learning as a broader educational paradigm is defined as a model combining face-to-face and online instruction, also termed blended learning." |
| [16] | Myers and Schenkman (2017) | Physical Therapy students | " . . . defined as clinical learning experiences embedded within the didactic curriculum, developed in collaboration with multiple stakeholders." |
| [23] | Baker et al., (2017) | Medical students | " . . . basic science knowledge is contextualized within the types of clinical presentations and diagnoses that students are likely to encounter during clerkships or residency." |
| [25] | Mawdsley and Willis (2018) | Pharmacy students | "An integrated curriculum is conceptualized as producing graduates who can understand, evaluate, and manage patients with complex drug regimens by drawing on a solid foundation in the basic and clinical sciences as applied to practice."<br>"Integration, then, was conceptualized as offering learners scaffolds to connect new meaning between knowledge and showing learners how to construct new understandings through vertical and horizontal integration of knowledge, connecting theory to its practical application." |

**Table A4.** *Cont.*

| Reference | Authors/Year | Learners | Authors' Explicit Definition of Integrated Learning, Education, Curriculum, or Teaching |
|---|---|---|---|
| [26] | Akram et al., (2018) | Medical students | "An integrated curriculum establishes significant linkages between the subjects or skills . . . Moreover, it allows opportunities for all the stakeholders to think outside the box." |
| [27] | Atta and AlQahtani (2018) | Medical students | " . . . holistic advance wherein the basic sciences are being delivered as a compound of the disciplines . . . with clinical perception from the early academic phase in a horizontally integrated manner." "Furthermore, the four major clinical sciences are considered in the teaching of clinical phases in the pre-graduation years of the curriculum in a vertically integrated manner." "Another form of integration is the spiral form which is defined as a curriculum involving, 'learning basic and clinical sciences crosswise', where, 'both theme and time matter.'" |
| [31] | Zumwalt and Dominguez (2019) | Advanced PhD basic science trainees planning on medical education careers. | "An integrated medical curriculum, defined as one where the connections between basic sciences and clinical sciences are highlighted and emphasized." |
| [33] | Mawdsley and Willis (2019) | Pharmacy students | "An integrated curriculum is one designed to provide learners with opportunities to create connections between knowledge, and to respond to problems by combining knowledge and skills from different disciplines to facilitate higher-order integrative learning." |
| [10] | Jette et al., (2020) | Physical therapy students | "Integrated learning can be separated into three categories- horizontal, vertical, and spiral." "Integration is more than the structure of the curriculum but reflects the process of learning as students develop understanding of how concepts fit together." |
| [36] | Kapitonova et al., (2020) | Medical students | "For a long time, medical education was focused on the acquisition of professional knowledge and skills, while currently this approach may no longer be considered sufficient, as modern graduates also require an ability to communicate, collaborate, develop logical constructions and obtain the skills to do research and conduct scientific discussions. In various universities in the world, these aspects are integrated into the goals of educational programs." |
| [11] | Hendriksen et al., (2020) | Pharmacy students | "Integrated learning" involves near-peer teaching of senior students training junior students, and can be focused on experiential learning." |
| [40] | Lee et al., (2020) | Dental students | "Integrated learning was defined through the Medical/Dental Humanities-Social Medicine/Dentistry(MDHS) education rubric. MDHS is an interdisciplinary approach to medical/dental education that seeks to incorporate relevant learning experiences in the humanities and social sciences into medicine and dentistry." |

**Table A4.** *Cont.*

| Reference | Authors/Year | Learners | Authors' Explicit Definition of Integrated Learning, Education, Curriculum, or Teaching |
|---|---|---|---|
| [45] | Quesnelle et al., (2021) | Medical students | "A restructuring of medical education curricula in which basic science disciplines are combined with organ-system blocks of instruction as longitudinal threads within the pre-clerkship curriculum." |
| [46] | Parrish et al., (2021) | Graduate students seeking initial secondary teacher certification | "... a model that integrates both [online asynchronous and synchronous] modes of engagement." |

**Table A5.** Summary of the implied definitions of integrated learning, education, curriculum, or teaching. N/S = none specified.

| Reference | Authors/Year | Learners | Authors' Implied Definition of Integrated Learning, Education, Curriculum, or Teaching |
|---|---|---|---|
| [21] | Corvin et al., (2017) | Masters of Public Health students | No formal definition given. Corvin et al., (2017) loosely allude to their integrated curriculum being one that exposes students to principles, theories, and constructs in a cross-disciplinary way, and include knowledge-based content and application of that knowledge in a longitudinal manner. |
| [24] | Carvour et al., (2018) | Medical students | No formal definition given. Carvour et al., (2018) point to "clinical integration" as involving assimilation of information and patient-centered application (e.g., communication with patient and team, dealing with uncertainty, lifelong learning). |
| [29] | Gustin et al., (2018) | Medical students | No formal definition given. Gustin et al., (2018) allude to breaking down the barriers between basic and clinical science. |
| [35] | Fatima et al., (2020) | Medical students | No formal definition given. Fatima et al., (2020) reference a paper with the following definition for integrated curriculum: education that is organized in such a way that it cuts across subject matter lines, bringing together various aspects of the curriculum into meaningful association to focus upon broad areas of study. |
| [42] | Strömbergsson et al., (2020) | Speech and Language Pathology students | No formal definition given. Strömbergsson et al., (2020) imply that integrated learning is defined as utilizing both vertical integration (linking earlier and later courses) and horizontal integration (and linking different subjects at the same time) models. |
| [43] | Abu Farha et al., (2021) | Clinical Nutrition and Dietetics students | No formal definition given. Abu Farha et al., (2021) imply that learning methods focus on educating students on how to learn actively and independently. |

**Table A6.** Summary of the educationally relevant theories or frameworks cited to guide development of integrated learning, education, curriculum, or teaching. N/S = none specified.

| Reference | Authors/Year | Learners | Educational Theories or Frameworks Cited to Guide Development of Integrated Learning, Education, Curriculum, or Teaching |
|---|---|---|---|
| [14] | Bernauer and Fuller (2017) | N/S | Constructivism, Blended learning. |
| [15] | Picciano (2017) | N/S | Bloom's taxonomy, Gagne's cognitivist instructional design, community of inquiry framework, connectivism, online collaborative learning, blending with pedagogical purpose model, multimodal model for online education. |
| [16] | Myers and Schenkman (2017) | Physical Therapy students | Kern's six steps for curriculum development. |
| [19] | Holthauser et al., (2017) | Medical students | The Association of American Medical Colleges (AAMC) Advisory Committee on Sexual Orientation, Gender Identity, and Sex Development issued a competency-based report, "Implementing Curricular and Institutional Climate Changes to Improve Health Care for Individuals who are LGBT, Gender Nonconforming, or Born with DSD: A Resource for Medical Educators" was used as a framework. Glassick's educational scholarship criteria. |
| [20] | DeBate et al., (2017) | Masters of Public Health students | Intervention mapping framework. |
| [21] | Corvin et al., (2017) | Masters of Public Health students | Intervention mapping framework. |
| [26] | Akram et al., (2018) | Medical students | Bloom's taxonomy: cognitive, psychomotor, affective domains. |
| [29] | Cahn et al., (2018) | Medical, nursing, occupational therapy, physical therapy, physician assistant, and speech-language pathology students | "A theoretical model from the management literature can help frame how to integrate the core competencies for interprofessional practice with uni-professional curricula . . . educators need to provide multiple settings and configurations for learners to hone their collaborative skills so that they become an unquestioned part of their practice." |
| [30] | Gustin et al., (2018) | Medical students | Constructivism. |
| [34] | McIlwaine et al., (2019) | Dental students, dental therapy and hygiene students | Integrated curriculum content was aligned to markers of quality inter-professional education as set out by Thistlethwaite and Nisbet, 2000 [52]. |
| [10] | Jette et al., (2020) | Physical therapy students | Cognitive load theory, constructivism. |
| [36] | Kapitonova et al., (2020) | Medical students | Spiral curriculum. |
| [38] | Hendriks et al., (2020) | Medical students | Self-determination theory, self-regulated learning theory. |
| [40] | Lee et al., (2020) | Dental students | The rubric of Medical/Dental Humanities-Social Medicine/Dentistry (MDHS) education. |
| [42] | Strömbergsson et al., (2020) | Speech and Language Pathology students | Horizontal integration, vertical integration, spiral curriculum. |
| [43] | Abu Farha et al., (2021) | Clinical Nutrition and Dietetics students | The nutrition care process (NCP), vertical integration. |
| [44] | Malhotra et al., (2021) | Pharmacy, Medicine, Psychology, and Nursing students | Cognitive flexibility theory, constructivism. |

**Table A6.** *Cont.*

| Reference | Authors/Year | Learners | Educational Theories or Frameworks Cited to Guide Development of Integrated Learning, Education, Curriculum, or Teaching |
|---|---|---|---|
| [45] | Quesnelle et al., (2021) | Medical students | Interactive, constructive, active, passive (ICAP) framework. |
| [46] | Parrish et al., (2021) | Graduate students seeking initial secondary teacher certification | Community of inquiry framework. |

**Table A7.** Summary of the actional educational practices intended to achieve integrated learning, education, curriculum, or teaching. N/S = none specified.

| Reference | Authors/Year | Learners | Actionable Educational Practices Intended to Achieve Integrated Learning, Education, Curriculum, or Teaching |
|---|---|---|---|
| [14] | Bernauer and Fuller (2017) | N/S | Integrated assessment, flipped classroom. |
| [16] | Myers and Schenkman (2017) | Physical Therapy students | Planned integrated clinical experiences, skill check-offs. |
| [17] | Moran Tovin et al., (2017) | Physical Therapy students | Classroom instruction, activities/labs. |
| [18] | Sharma et al., (2017) | Medical students | Integrated co-teaching (e.g., biochemistry and general medicine: diabetes mellitus and alcohol and liver disease). |
| [19] | Holthauser et al., (2017) | Medical students | Including inclusive and affirmative content addressing disparities as part of the daily practice of caring for all patients, and content specifically tailored for lesbian, gay, bisexual, transgender, GNC, or DSD populations. Lecture, problem-based learning (PBL), standardized patients, small-group discussion, reflective writing, patient panel. |
| [21] | Corvin et al., (2017) | Masters of Public Health students | Flipped classroom including pre-event work, video lectures, in-class practical exercises, and supplemental reading), problem-based learning (PBL). |
| [22] | Tshibwabwa et al., (2017) | Medical students | Problem-based learning (PBL), Qstream (online spaced education software). |
| [23] | Baker et al., (2017) | Medical students | Embed basic science topics within clinical problems, problem-based learning (PBL), task-based learning, case-based learning; drill downs. |
| [24] | Carvour et al., (2018) | Medical students | Student-directed online learning activities, large group teaching, team-based learning. |
| [26] | Akram et al., (2018) | Medical students | Clinical skills are chained with basic medical sciences through simulated skills in arranged topics, thus, cognitive and psychomotor domains are combined while affective domain is placed vertically. Lecture, seminar, tutorial, problem-based learning (PBL), case-based learning, bedside teaching. |
| [27] | Atta and AlQahtani (2018) | Medical students | Practical laboratory sessions, problem-based learning (PBL), self-directed learning, seminars, field visits, bedside teaching, clinical case orientation, skill laboratory sessions. |

**Table A7.** *Cont.*

| Reference | Authors/Year | Learners | Actionable Educational Practices Intended to Achieve Integrated Learning, Education, Curriculum, or Teaching |
|---|---|---|---|
| [28] | Yue et al., (2018) | Medical students | PBL, lecture. |
| [29] | Cahn et al., (2018) | Medical, nursing, occupational therapy, physical therapy, physician assistant, and speech-language pathology students | Classroom (observed live patient case rounds conducted by experienced clinicians), simulation (interprofessional education events involving activities with standardized patients) Community (a common book event for all programs with discussion facilitated by an interprofessional team of faculty members. Service-learning day with a local non-profit organization and subsequent reflection). Team (using on-campus, pro-bono health centers "designed to provide interprofessional care to clients and interprofessional learning to students"). |
| [30] | Gustin et al., (2018) | Medical students | Lectures, problem-based learning (PBL). |
| [31] | Zumwalt and Dominguez (2019) | Advanced PhD basic science trainees planning on medical education careers. | Didactics, clinical shadowing. |
| [33] | Mawdsley and Willis (2019) | Pharmacy students | Integrated Professional Practice (IPP): a core of work placements and professional practice learning. |
| [34] | McIlwaine et al., (2019) | Dental students, dental therapy and hygiene students | Enquiry-based learning (EBL), supported by plenaries, workshops, and self-directed learning. Simulated dental learning environment. |
| [10] | Jette et al., (2020) | Physical therapy students | Case-based learning. |
| [32] | Kapitonova et al., (2020) | Medical students | Problem-based learning (PBL); progress tests and quizzes for each organ system that use multiple choice questions (MCQs), single answer questions (SAQs), single best answer questions (SBAQ), objective structured practical examination (OSPE), problem-based questions (PBQ), modified essay questions (MEQ). |
| [37] | Gergen et al., (2020) | Medical students | Small group chalk talk, peer teaching, whiteboard lectures. |
| [38] | Hendriks et al., (2020) | Medical students | Massive open online courses (MOOCs). |
| [11] | Hendriksen et al., (2020) | Pharmacy students | Peer teaching, standardized patients, team-based development, problem-based learning. |
| [39] | Banning et al., (2020) | Physician Assistant students | Art-viewing and reflective question prompts. |
| [41] | Venkatesh et al., (2020) | Medical students | Online interactive videos with built-in questions. Videos contained clinical case presentations, images, digital microscopy slides and laboratory reports. A 1 h synthesizing session, which involved an integrated lecture-based review of the learning activities was provided to the students with a question/answer and discussion time. |
| [43] | Abu Farha et al., (2021) | Clinical Nutrition and Dietetics students | Case-based learning (CBL). |
| [44] | Malhotra et al., (2021) | Pharmacy, Medicine, Psychology, and Nursing students | Interprofessional education: lectures, high fidelity simulation lab, case conference, grand rounds. |
| [46] | Parrish et al., (2021) | Graduate students seeking initial secondary teacher certification | Flipped classroom, multiple-choice pre-event quizzes, team-based learning, application activities. |

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
