# Peer review of "A Systematic Review of Integrated Learning Definitions, Frameworks, and Practices in Recent Health Professions Education Literature"

_education, doi:10.3390/educsci12030165_

Round 1

Reviewer 1 Report

Dear authors,

I think you are covering a very important topic in healthcare education and research and education would benefit from a more elaborate use of the term "integration". However, I see some limitations in your review study and have some suggestions below, which I hope you might find helpful.  

Introduction:

  • It is a bit confusing that you start with discipline-related integration, but in my view this is only an example fo forms of integration. Others could be integration of different types of teaching methods, different health professions etc. So, for me it is unlcear whether you see the discipline-related integration as an example or this is the (only) form of integration you are focusing on with your review. It would help to clarify this.
  • Also, the research question seems more general on integration than integrated learning?
  • The purpose of your study could be based more in existing literature on integration frameworks (that are mentioned later in the discussion). This might help the reader to better understand the current staus concerning integration. 

Methods:

  • Why did you limit the search time to 2017-21 and not go further back?
  • Why did you not create a more sophisticated search term with your inclusion criteria? Your search query seems very general, but on the other hand would not find articles that use "integration" instead of "integrated". Was this on purpose - but you are not only discussing "integrated curriculum" etc. but also "integration", so I am a bit confused about the exact focus of your review.
  • Who and how did you do the screening of the articles and decide about inclusion/exclusion?  Who and how did you do the fulltext review? What kind of quality criteria / protocols did you apply?

Results:

  • It is a bit cumbersome to identify the results that refer to your research questions in  the long tables. I would suggest to move large parts into an appendix and focus on describing the relevant (based on your research questions) results (see also my comments for discussion)

Discussion:

  • In the first paragraphs results are introduced (percentages of articles), I would suggest showing these in the results section.
  • Also later on, the themes are discussed, but not presented in the results section (I would see the themes as a result).
  • Existing frameworks (e.g. ICAP) could be introduced already in the introduction.

Author Response

Thank you for your comments and feedback, and I wish to acknowledge the value of your contributions to improving this article. My co-authors and I are grateful for peer-reviewers who contribute to assuring that scholarship entering the field is of high quality and grounded in previous findings and appropriate methodologies.

Reviewer 2 Report

The considerations presented in the paper constitute a very interesting and original study of the subject matter related to education for medical professions. The purpose of this systematic review was to explore recent health professions education scholarship concerning reported definitions, theories or conceptual frameworks, and actionable educational activities around integrated learning, education, curricula, or teaching. The goal has been achieved.

Author Response

(The authors gave the same response as above.)

Reviewer 3 Report

The overall manuscript is neat and written concisely—with relevant information for existing literature. The formal way of writing is a lovely read. Also the overall neatness of the manuscript deserves compliments. Improvements can be made by making the results easier to read by moving the tables to the Appendix and set up concise paragraphs listing main results. This also means that information in the discussion can be moved to the results section, especially with the examples you list (those are results/examples from your sample). Your discussion is to place those results in a broader context.

Author Response

(The authors gave the same response as above.)

Round 2

Reviewer 3 Report

Overall sufficient revisions. Recheck them for double periods and other punctuation/ typing/ spelling errors. I am still unsure about the phrasing of your RQ (you might or might not change that although I would advise you to do so): 

  1. Do authors define integrated learning, integrated education, integrated curriculum, or integrated teaching, and if so what do they mean? = How do authors define integrated learning, integrated education, integrated curriculum, or integrated teaching?
  2. What theories or conceptual frameworks do authors turn to for guidance in developing integrated learning, education, curriculum, or teaching? = What theories or conceptual frameworks are used as guidance in developing in integrated learning, education, curriculum, or teaching?
  3. What have authors done practically to implement integrated learning, education, curriculum, or teaching among their learners? = How can practitioners integrate learning, education, curriculum, or teaching? The addition of "among their learners" is redundant. I do not understand this addition. 

Author Response

Thank you again for investing your time to read and review our article.
